# Contrastive Learning-based Sentence Encoders Implicitly Weight Informative Words

**Hiroto Kurita**[1]    **Goro Kobayashi**[1,2]    **Sho Yokoi**[1,2]    **Kentaro Inui**[3,1,2]

[1] Tohoku University    [2] RIKEN    [3] MBZUAI

{hiroto.kurita, goro.koba}@dc.tohoku.ac.jp

yokoi@tohoku.ac.jp  kentaro.inui@mbzuai.ac.ae

## Abstract

The performance of sentence encoders can be significantly improved through the simple practice of fine-tuning using contrastive loss. A natural question arises: what characteristics do models acquire during contrastive learning? This paper theoretically and experimentally shows that contrastive-based sentence encoders implicitly weight words based on information-theoretic quantities; that is, more informative words receive greater weight, while others receive less. The theory states that, in the lower bound of the optimal value of the contrastive learning objective, the norm of word embedding reflects the information gain associated with the distribution of surrounding words. We also conduct comprehensive experiments using various models, multiple datasets, two methods to measure the implicit weighting of models (Integrated Gradients and SHAP), and two information-theoretic quantities (information gain and self-information). The results provide empirical evidence that contrastive fine-tuning emphasizes informative words.

https://github.com/kuriyan1204/
sentence-encoder-word-weighting

## 1 Introduction

Embedding a sentence into a point in a high-dimensional continuous space plays a foundational role in the natural language processing (NLP) (Arora et al., 2017; Reimers and Gurevych, 2019; Chuang et al., 2022, etc.). Such sentence embedding methods can also embed text of various types and lengths, such as queries, passages, and paragraphs; therefore, they are widely used in diverse applications such as information retrieval (Karpukhin et al., 2020; Muennighoff, 2022), question answering (Nguyen et al., 2022), and retrieval-augmented generation (Chase, 2023).

One of the earliest successful sentence embedding methods is additive composition (Mitchell and Lapata, 2010; Mikolov et al., 2013), which embeds a sentence (i.e., a sequence of words)

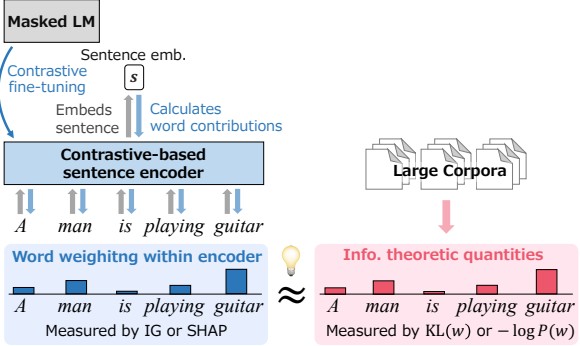

Figure 1: Overview of our study. We quantify the word weighting within contrastive-based sentence encoders by XAI techniques: Integrated Gradients (IG) or Shapley additive explanations (SHAP). We found that the quantified weightings are close to infromation-theoretic quantities: information-gain KL(w) and self-information $-\log P(w)$.

by summing its static word embeddings (**SWEs**; Mikolov et al., 2013; Pennington et al., 2014). Besides, *weighing each word based on the inverse of word frequency* considerably improved the quality of the sentence embeddings, exemplified by TF-IDF (Arroyo-Fernández et al., 2019) and smoothed inverse word frequency (SIF; Arora et al., 2017).

Recent sentence embeddings are built on masked language models (**MLMs**; Devlin et al., 2019; Liu et al., 2019; Song et al., 2020). Although sentence embeddings from the additive composition of MLMs' word embeddings are inferior to those of SWEs (Reimers and Gurevych, 2019), fine-tuning MLMs with contrastive learning objectives has elevated the quality of sentence embeddings (Reimers and Gurevych, 2019; Gao et al., 2021; Chuang et al., 2022, etc.) and is now the de-facto standard. Interestingly, these contrastive-based sentence encoders do *not employ explicit word weighting*, which is the key in the SWE-based methods.

In this paper, we demonstrate that a reason for the success of contrastive-based sentence encoders is *implicit word weighting*. Specifically, by com-

bining explainable AI (XAI) techniques and information theory, we demonstrate that *contrastive-based sentence encoders implicitly weight each word according to two information-theoretic quantities* (Figure 1). To measure the contribution (i.e., implicit weighting) of each input word to the output sentence embedding within the encoders, we used two XAI techniques: Integrated Gradients (IG; Sundararajan et al., 2017) and Shapley additive explanations (SHAP; Lundberg and Lee, 2017) (Section 3.1). To measure the information-theoretic quantities of each word, we used the two simplest quantities, information-gain $\mathrm{KL}(w)$ and self-information $-\log P(w)$ (Section 3.2). To demonstrate our hypothesis, we first provided a theoretical connection between contrastive learning and information gain (Section 4). We then conducted comprehensive experiments with a total of 12 models and 4 datasets, which found a strong empirical correlation between the encoders' implicit word weighting and the information-theoretic quantities (Section 5). The results of our study provide a bridge between SWE-era explicit word weighting techniques and the implicit word weighting used by recent contrastive-based sentence encoders.

## 2 Contrastive-Based Sentence Encoders

This section provides an overview of contrastive-based sentence encoders such as SBERT (Reimers and Gurevych, 2019) [1] and SimCSE (Gao et al., 2021). These models are built by fine-tuning MLMs with contrastive learning objectives.

**Input and Output:** As shown in Figure 1, a contrastive-based sentence encoder $\boldsymbol{m}\colon \mathbb{R}^{n\times d} \to \mathbb{R}^d$ calculates a sentence embedding $\boldsymbol{s} \in \mathbb{R}^d$ from a sequence of input word embeddings $\boldsymbol{W} = [\boldsymbol{w}_1, \ldots, \boldsymbol{w}_n] \in \mathbb{R}^{n\times d}$ corresponding to input words and special tokens. For example, for the sentence "*I like playing the piano*," the input to the encoder $\boldsymbol{m}$ is $\boldsymbol{W} = [\boldsymbol{w}_{\texttt{[CLS]}}, \boldsymbol{w}_I, \boldsymbol{w}_{like}, \ldots, \boldsymbol{w}_{\texttt{[SEP]}}]$, and sentence embedding $\boldsymbol{s}$ is calculated as follows:

$$\boldsymbol{s} := \boldsymbol{m}(\boldsymbol{W}) = [m_1(\boldsymbol{W}), \ldots, m_d(\boldsymbol{W})], \quad (1)$$

where $m_i\colon \mathbb{R}^{n\times d} \to \mathbb{R}$ denotes the computation of the $i$-th element of $\boldsymbol{s}$.

**Architecture:** The encoder architecture consists of (i) Transformer layers (same as MLM architecture) and (ii) a pooling layer. (i) Transformer layers

---

update input word embeddings to contextualized word embeddings $\boldsymbol{w}_i \mapsto \boldsymbol{e}_i$. (ii) The pooling layer then pools the $n$ representations $[\boldsymbol{e}_1, \ldots, \boldsymbol{e}_n]$ into a single sentence embedding $\boldsymbol{s} \in \mathbb{R}^d$. There are two major pooling methods, MEAN and CLS: MEAN averages the contextualized word embeddings, and CLS just uses the embeddings for a [CLS] token after applying an MLP on top of it.

**Contrastive fine-tuning:** The contrastive fine-tuning of MLMs is briefly described below. In contrastive learning, positive pairs $(\boldsymbol{s}, \boldsymbol{s}_{\mathrm{pos}})$, i.e., semantically similar pairs of sentence embeddings, are brought closer, while negative pairs $(\boldsymbol{s}, \boldsymbol{s}_{\mathrm{neg}})$ are pushed apart in the embedding space. Positive examples $\boldsymbol{s}_{\mathrm{pos}}$ and negative examples $\boldsymbol{s}_{\mathrm{neg}}$ for a sentence embedding $\boldsymbol{s}$ are created in different ways depending on the method. For instance, in the unsupervised SimCSE (Gao et al., 2021), a positive example $\boldsymbol{s}_{\mathrm{pos}}$ is created by embedding the same sentence as $\boldsymbol{s}$ by applying a different dropout; and a negative example $\boldsymbol{s}_{\mathrm{neg}}$ is created by embedding a sentence randomly sampled from a training corpus.

## 3 Analysis Method

We compare the implicit word weighting within the contrastive-based sentence encoders with information-theoretic quantities of words. Here we introduce (i) quantification of the implicit word weighting within the encoders using two XAI techniques (Section 3.1) and (ii) two information-theoretic quantities of words (Section 3.2).

### 3.1 Implicit Word Weighting within Encoder

Contrastive-based sentence encoders are not given explicit word weighting externally but are expected to implicitly *weight* words through the complicated internal network. We quantify the implicit word weighting using two widely used feature attribution methods (Molnar, 2022): Integrated gradients (Sundararajan et al., 2017) and Shapley additive explanations (Lundberg and Lee, 2017).

### 3.1.1 Integrated Gradients (IG)

Integrated Gradients (**IG**) is a widely XAI technique used to calculate the contribution of each input feature to the output in neural models. IG has two major advantages: (i) it is based on the gradient calculations and thus can be used to arbitrary neural models; and (ii) it satisfies several desirable properties, for example, the sum of the contributions for

---

[1]A classification loss is used in the training of SBERT; however, this can be roughly regarded as a contrastive loss (see Appendix A).

each input feature matches the output value (*Completeness* described in Sundararajan et al., 2017). It has also been actively applied to the analysis of MLM-based models (Hao et al., 2021; Prasad et al., 2021; Bastings et al., 2022; Kobayashi et al., 2023).

The formal definition of IG is as follows: Let $f\colon \mathbb{R}^{n\times d} \to \mathbb{R}$ be a model (e.g., each element of sentence encoder) and $\boldsymbol{X}' \in \mathbb{R}^{n\times d}$ be a certain input (e.g., word vectors). IG calculates a contribution score $\mathrm{IG}_{i,j}$ for the $(i,j)$ element of the input $\boldsymbol{X}'[i,j]$ (e.g., each element of each input word vector) to the output $f(\boldsymbol{X}')$:

$$f(\boldsymbol{X}') = \sum_{i=1}^{n}\sum_{j=1}^{d} \mathrm{IG}_{i,j}(\boldsymbol{X}';f,\boldsymbol{B}) + f(\boldsymbol{B}) \quad (2)$$

$$\mathrm{IG}_{i,j}(\boldsymbol{X}';f,\boldsymbol{B}) \coloneqq (\boldsymbol{X}'[i,j] - \boldsymbol{B}[i,j])$$
$$\times \int_{\alpha=0}^{1} \frac{\partial f}{\partial \boldsymbol{X}[i,j]}\bigg|_{\boldsymbol{X}=\boldsymbol{B}+\alpha(\boldsymbol{X}'-\boldsymbol{B})} \mathrm{d}\alpha. \quad (3)$$

Here $\boldsymbol{B}$ denotes a baseline vector, often an uninformative or neutral input is employed. Notably, IG decomposes the output value into the sum of the contribution scores of each input (Equation 2).

**Application to the sentence encoder:** We aim to measure the contribution of each input word to the output sentence embedding. However, when applying IG to the sentence encoder $\boldsymbol{m}$ (its $k$-th element is $m_k$) and input $\boldsymbol{W} = [\boldsymbol{w}_1,\ldots,\boldsymbol{w}_n]$, it can only compute the fine contribution score of the $j$-th element of the each input word vector $\boldsymbol{w}_i$ to the $k$-th element of the sentence vector $\boldsymbol{s} = \boldsymbol{m}(\boldsymbol{W})$. Thus, we aggregates the contribution scores across all the $(j,k)$ pairs by the Frobenius norm:

$$c_i(\boldsymbol{W};\boldsymbol{m},\boldsymbol{B}) \coloneqq \sqrt{\sum_{j=1}^{d}\sum_{k=1}^{d} \mathrm{IG}_{i,j}(\boldsymbol{W}';m_k,\boldsymbol{B})^2},$$
$$\boldsymbol{B} = [\boldsymbol{w}_{\mathrm{CLS}}, \boldsymbol{w}_{\mathrm{PAD}}, \boldsymbol{w}_{\mathrm{PAD}}, \ldots, \boldsymbol{w}_{\mathrm{PAD}}, \boldsymbol{w}_{\mathrm{SEP}}], \quad (4)$$

where we used a sequence of input word vectors for an uninformed sentence "[CLS] [PAD] [PAD] . . . [PAD] [SEP]" as the baseline input $\boldsymbol{B}$. In addition, the word contribution $c_i$ is normalized with respect to the sentence length $n$ to compare contributions equally among words in sentences of different lengths: $c'_i \coloneqq c_i/(\frac{1}{n}\sum_{j=1}^{n} c_j)$.

### 3.1.2 SHapley Additive exPlanations (SHAP)

Shapley additive explanations (**SHAP**) is a feature attribution method based on Shapley values (Shapley, 1953). Similar to IG, SHAP satisfies the desirable property: it linearly decomposes the model

output to the contribution of each input (Lundberg and Lee, 2017). Its formal definition and application to the word weighting calculation of contrastive-based sentence encoders are shown in Appendix D.

Though we can apply SHAP to analyze sentence encoders, SHAP is often claimed to be unreliable (Prasad et al., 2021). Thus, we discuss the experimental results using IG in the main text and show the results using SHAP in Appendix E.

### 3.2 Information-Theoritic Quantities

Here, we introduce two information-theoretic quantities that represent the amount of information a word conveys.

### 3.2.1 Information Gain KL($w$)

The first quantity is the information gain, which measures how much a probability distribution (e.g., the unigram distribution in some sentences) changes after observing a certain event (e.g., a certain word). Information gain of observing a word $w$ in a sentence is denoted as $\mathrm{KL}(w) \coloneqq \mathrm{KL}(P_{\mathrm{sent}}(\cdot|w) \parallel P(\cdot))$, where $P_{\mathrm{sent}}(\cdot|w)$ is the word frequency distribution in sentences containing word $w$, and $P(\cdot)$ is a distribution without conditions. Intuitively, $\mathrm{KL}(w)$ represents the extent to which the topic of a sentence is determined by observing a word $w$ in the sentence. For example, if $w$ is "*the*", $\mathrm{KL}("the")$ becomes small because the information that a sentence contains "*the*" does not largely change the word frequency distribution in the sentence from the unconditional distribution ($P_{\mathrm{sent}}(\cdot|"the") \approx P(\cdot)$). On the other hand, if $w$ is "*NLP*", $\mathrm{KL}("NLP")$ becomes much larger than $\mathrm{KL}("the")$ because the information that a sentence contains "*NLP*" is expected to significantly change the word frequency distribution ($P_{\mathrm{sent}}(\cdot|"NLP") \neq P(\cdot)$). Recently, Oyama et al. (2023) showed that $\mathrm{KL}(w)$ is encoded in the norm of SWE. Also, $\chi^2$-measure, which is a similar quantity to $\mathrm{KL}(w)$, is useful in keyword extraction (Matsuo and Ishizuka, 2004). We provide a theoretical connection between $\mathrm{KL}(w)$ and contrastive learning in Section 4.

### 3.2.2 Self-Information $-\log P(w)$

The second quantity which naturally represents the information of a word is self-information $-\log P(w)$. $-\log P(w)$ is based on the inverse of word frequency and actually very similar to word weighting techniques used in SWE-based sen-

tence encoding methods such as TF-IDF ([Arroyo-Fernández et al., 2019](#)) and SIF weighting ([Arora et al., 2017](#)). Note that the information gain $\mathrm{KL}(w)$ introduced in Section 3.2.1 is also correlated with the inverse of word frequency ([Oyama et al., 2023](#)), and both quantities introduced in this section are close to the word weighting techniques. Detailed comparison between the two quantities and the word weighting techniques is shown in Appendix B. If the contrastive-based sentence encoder's word weighting is close to $\mathrm{KL}(w)$ and $-\log P(w)$, the post-hoc word weighing used in SWE-based methods is implicitly learned via contrastive learning.

## 4 Theoretical Analysiss

This section provides brief explanations of the theoretical relationship between $\mathrm{KL}(w)$ and contrastive learning. Given a pair of sentences $(s, s')$, contrastive learning can be regarded as a problem of discriminating whether a sentence $s'$ is a positive (semantically similar) or negative (not similar) example of another sentence $s$. We reframe this discrimination problem using word frequency distribution. After observing $w$ in $s$, the *positive example* is likely to contain words that co-occur with $w$; i.e., the word distribution of the positive example likely follows $P_{\mathrm{sent}}(\cdot|w)$. Contrary, the *negative example* is likely to contain random words from a corpus regardless of the observation of $w$; i.e., the word distribution of the negative example likely follows $P(\cdot)$. Hence, $\mathrm{KL}(P_{\mathrm{sent}}(\cdot|w) \parallel (P(\cdot)) = \mathrm{KL}(w)$ approximately represents the objective of the discrimination problem (i.e., contrastive learning). See Appendix C for a more formal explanation.

## 5 Experiments

In this section, we investigate the empirical relationship between the implicit word weighting within contrastive-based sentence encoders (quantified by IG or SHAP) and the information-theoretic quantities ($\mathrm{KL}(w)$ or $\log P(w)$).

### 5.1 Experimental Setup

**Models:** We used the following 9 sentence encoders: SBERT ([Reimers and Gurevych, 2019](#)), Unsupervised/Supervised SimCSE ([Gao et al., 2021](#)), DiffCSE ([Chuang et al., 2022](#)), both for BERT and RoBERTa-based versions, and `all-mpnet-base-v2`[2]. As baselines, we also

[2] https://huggingface.co/sentence-transformers/all-mpnet-base-v2

| Model (pooling) | $R^2 \times 100 \uparrow$ | $\beta \times 100 \uparrow$ |
|---|---|---|
| BERT (CLS) | 0.0 | 0.2 |
| → U.SimCSE-BERT | **32.4** (+32.4) | **17.3** (+17.1) |
| → S.SimCSE-BERT | **30.8** (+30.8) | **16.9** (+16.7) |
| → DiffCSE-BERT | **31.6** (+31.6) | **13.1** (+12.9) |
| BERT (MEAN) | 24.4 | 7.9 |
| → SBERT | 22.9 (−1.5) | **16.1** (+8.1) |
| RoBERTa (CLS) | 2.0 | 5.0 |
| → U.SimCSE-RoBERTa | **29.7** (+27.7) | **17.5** (+12.5) |
| → S.SimCSE-RoBERTa | **29.1** (+27.1) | **20.2** (+15.2) |
| → DiffCSE-RoBERTa | **28.4** (+26.4) | **16.5** (+11.4) |
| RoBERTa (MEAN) | 4.2 | 5.5 |
| → SRoBERTa | **31.4** (+27.2) | **24.0** (+18.4) |
| MPNet (MEAN) | 0.6 | −2.6 |
| → all-mpnet-base-v2 | **21.6** (+21.0) | **21.0** (+23.5) |

Table 1: Coefficient of determination ($R^2$) and regression coefficient ($\beta$) of the linear regression of the words' weightings calculated by IG on the information gain KL for the STS-B dataset. The $R^2$ and $\beta$ is reported as $R^2 \times 100$ and $\beta \times 100$. The values inside the brackets represent the gain from pre-trained models.

analyzed 3 pre-trained MLMs: BERT ([Devlin et al., 2019](#)), RoBERTa ([Liu et al., 2019](#)) and MPNet ([Song et al., 2020](#)). All models are base size.

**Dataset:** We used STS-Benchmark ([Cer et al., 2017](#)), a widely used dataset to evaluate sentence representations, for input sentences to encoders and calculating $-\log P(w)$ and $\mathrm{KL}(w)$. We used the validation set, which includes 3,000 sentences. We also conducted experiments using Wikipedia, STS12 ([Agirre et al., 2012](#)), and NLI datasets ([Bowman et al., 2015](#); [Williams et al., 2018](#)) for the generalizability, which are shown in Appendix F.2.

**Experimental procedure:** First, we fed all sentences to the models and calculated the word weightings by IG or SHAP (Section 3.1). Then we applied OLS regression to the calculated word weightings on $-\log P(w)$ or $\mathrm{KL}(w)$ (Section 3.2) for each model. Although we experimented with all four possible combinations from the two XAI methods and two quantities for each model, we report here the results only for the combination of IG and $\mathrm{KL}(w)$. Other results are shown in Appendix E.

### 5.2 Quantitative Analysis

Table 1 lists the coefficient of determination ($R^2$) and regression coefficient ($\beta$) of the linear regression on IG and $\mathrm{KL}(w)$. Figure 2 shows the plots of the word weightings and their regression lines for BERT and Unsupervised SimCSE.

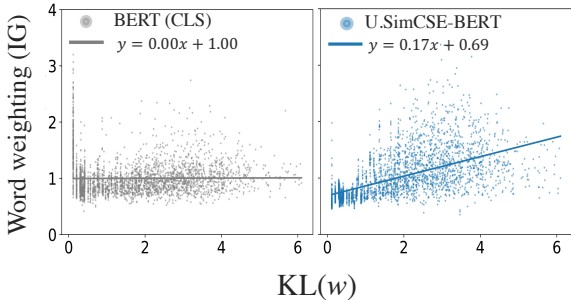

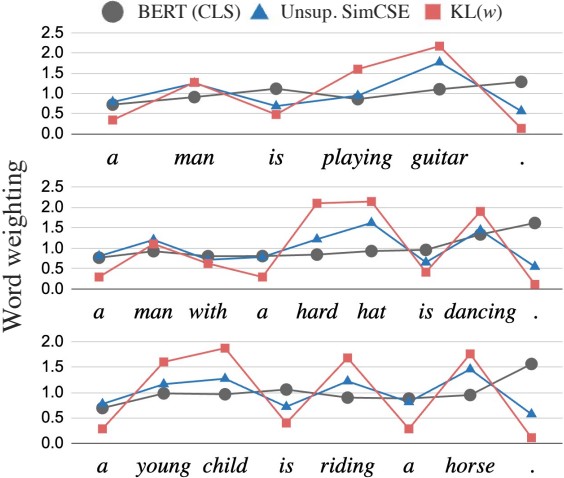

Figure 2: Linear regression plots between $\mathrm{KL}(w)$ and the word weighting of BERT (CLS) (left) and Unsupervised (U.) SimCSE-BERT (right) using IG. We plotted subsampled 3000 tokens from the tokens with the top 99.5% of small KL values for visibility. The plots $-\log P(w)$ experiments are shown in Figure 6 in Appendix E.

Figure 3: Sentence-level examples of the word weighting of BERT (CLS) and Unsupervised SimCSE-BERT using IG. Same as IG, $\mathrm{KL}(w)$ is normalized so that the sum of $\mathrm{KL}(w)$ becomes the sentence length for visibility. For words like *"man"*, *"child"*, *"guitar"*, *"is"*, *"hat"*, *"horse"* and *"."* the word weightings of the contrastive fine-tuned model (▲) are closer to $\mathrm{KL}(w)$ (■) than the pre-trained model (●), which means that contrastive fine-tuning induces word weighting by $\mathrm{KL}(w)$.

Table 1 shows that $R^2$ and $\beta$ are much higher in contrastive-based encoders than pre-trained models.[3] Similar trends were obtained with other XAI method, information-theoretic quantity, and datasets (Appendix E, F.2). These indicate that contrastive learning for sentence encoders induces word weighting according to the information-theoretic quantities (Figure 2). In other words, contrastive-based encoders acquired the inner mechanism to highlight informative words more. Furthermore, given that $\mathrm{KL}(w)$ and $\log P(w)$ are close to the weighting techniques employed in the SWE-based sentence embeddings (Section 3.2), these results suggest that the contrastive-based sentence encoders learn implicit weightings similar to the explicit ones of the SWE-based methods.

### 5.3 Qualitative Analysis

Figure 3 shows the word weighting within BERT (●) and Unsupervised SimCSE (▲) and $\mathrm{KL}(w)$ (■) for three sentences: *"a man is playing guitar."*, *"a man with a hard hat is dancing."*, and *"a young child is riding a horse."*. The contrastive-based encoder (Unsup. SimCSE; ▲) has more similar word weighting to $\mathrm{KL}(w)$ (■) than the MLM (BERT; ●), which is consistent with the $R^2$ in the Table 1. Also, the contrastive-based encoder (▲) tends to weight input words more extremely than the MLM (●), which is consistent with the $\beta$ in Table 1. For example, weights for nouns such as *"guitar"*, *"hat"*, *"child"*, and *"horse"* are enhanced, and weights for non-informative words such as *"is"*, and *"."* are

---

[3]Exceptionally, $R^2$ does not change much from BERT (MEAN) to SBERT, while $\beta$ indeed increases.

discounted by contrastive learning (● → ▲). On the other hand, weights for words such as *"a"* and *"with"*, whose $\mathrm{KL}(w)$ is very small, are not changed so much. Investigating the effect of POS on word weighting, which is not considered on $\mathrm{KL}(w)$ and $-\log P(w)$, is an interesting future direction.

## 6 Conclusion

We showed that contrastive learning-based sentence encoders implicitly weight informative words based on information-theoretic quantities. This indicates that the recent sentence encoders learn implicit weightings similar to the explicit ones used in the SWE-based methods. We also provided the theoretical proof that contrastive learning induces models to weight each word by $\mathrm{KL}(w)$. These provide insights into why contrastive-based sentence encoders succeed in a wide range of tasks, such as information retrieval (Muennighoff, 2022) and question answering (Nguyen et al., 2022), where emphasizing some informative words is effective. Besides sentence encoders, investigating the word weighting of retrieval models is an interesting future direction.

## Limitations

There are three limitations in this study. First is the limited experiment for baseline input of IG. For the IG experiment, we only tested the baseline input with PAD tokens explained in Section 3.1.1. Although there is no consensus on the appropriate baseline inputs of IG for contrastive-based sentence encoders, a comparison of results with different baseline inputs is left to future work. Second, our findings do not cover contrastive text embeddings from retrieval models. Analyzing contrastive-based retrieval models such as DPR (Karpukhin et al., 2020), Contriever (Izacard et al., 2022), and investigating the effect on text length would also be an interesting future direction. Third is the assumptions made in the sketch of proof in Section C. In our proof, we assume that the similarity of sentence embeddings is calculated via inner product. However, in practice, cosine similarity is often used instead. Also, we do not consider the contextualization effect of Transformer models on the word embeddings in the proof. The theoretical analysis using cosine similarity or considering the contextualization effect is left to future work.

## Ethics Statement

Our study showed that sentence encoders implicitly weight input words by frequency-based information-theoretic quantities. This suggests that sentence encoders can be affected by biases in the training datasets, and our finding is a step forward in developing trustful sentence embeddings without social or gender biases.

## Acknowledgements

We would like to thank the members of the Tohoku NLP Group for their insightful comments. This work was supported by JSPS KAKENHI Grant Number JP22J21492, JP22H05106; JST ACT-X Grant Number JPMJAX200S; and JST CREST Grant Number JPMJCR20D2, Japan.

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

## A The classification loss in SBERT/RoBERTa

SBERT and SRoBERTa (Reimers and Gurevych, 2019) are fine-tuned with softmax classifiers on top of original MLMs architectures. Before the softmax, the models calculate the element-wise difference of two sentence embeddings and concatenate them with the original two sentence embeddings. Then the concatenated embedding is linearly transformed and fed into a softmax classifier. Here, the element-wise difference is expected to work as an implicit similarity function. Thus, SBERT and SRoBERTa can be roughly regarded as contrastive-based sentence encoders.

## B Comparison of $\mathrm{KL}(w)$ and $-\log P(w)$ with Existing Weighting Methods

Here, we compare the two information-theoretic quantities $\mathrm{KL}(w)$ and $-\log P(w)$ with existing weighting methods used in SWE-based sentence embedding: TF-IDF (Arroyo-Fernández et al., 2019) and SIF-weighting (Arora et al., 2017). Figure 4 and 5 are the scatter plots of IDF and SIF-weighitng against $\mathrm{KL}(w)$ and $-\log P(w)$, respectively. We removed the term frequency part in TF-IDF and only used IDF to eliminate contextual effects and reduce the variance of the values for the comparison, and all quantities were calculated on the STS-Benchmark (Cer et al., 2017) dataset. From the figures, we can see that the two information-theoretic quantities $\mathrm{KL}(w)$ and

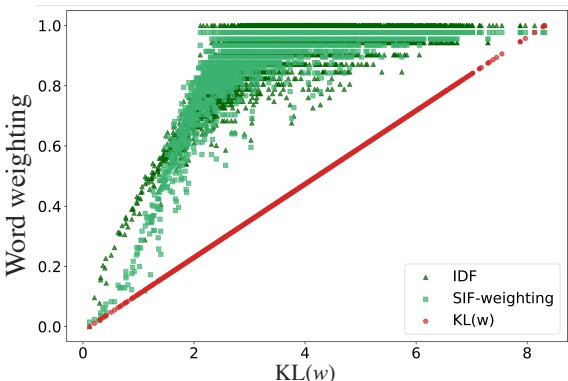

Figure 4: Scatter plot of $\mathrm{KL}(w)$, IDF, and SIF-Weighting. All weightings are normalized within the same weighting method.

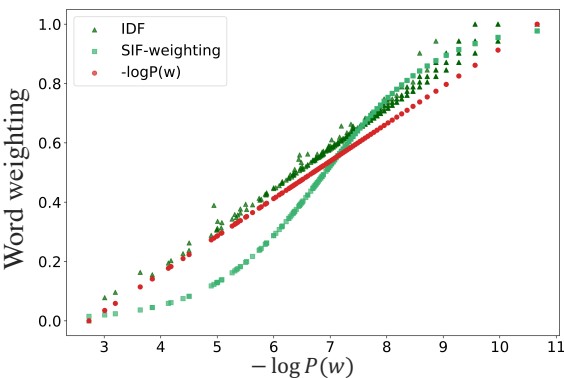

Figure 5: Scatter plot of $-\log P(w)$, IDF, and SIF-Weighting. All weightings are normalized within the same weighing method.

$-\log P(w)$ are highly correlated with existing word weighting TF-IDF (Arroyo-Fernández et al., 2019) and SIF-weighting (Arora et al., 2017).

## C The Theoretical Relationship between Contrastive Learning and Information Gain

Formally, the following theorem holds:

**Theorem 1.** *Let $\mathcal{S}$ be the set of sentences, $\{((s, s'), C)\}$ be the dataset constructed from $\mathcal{S}$ where $s \sim \mathcal{S}$, $s' = s$ (when $C = 1$), and $s' \sim \mathcal{S}$ (when $C = 0$). Suppose that the sentence encoder parametrized by $\theta$ takes a sentence $s = (w_1, \ldots, w_{|s|})$ as the input and returns word embeddings $(\boldsymbol{w}_1, \ldots, \boldsymbol{w}_{|s|})$ and its mean-pooled sentence embedding $\boldsymbol{s} = \frac{1}{|s|} \sum_{i=1}^{|s|} \boldsymbol{w}_i$ as the output, and the contrastive fine-tuning maximizes $\mathcal{L}_{\mathrm{contrastive}}(\theta)$, the log-likelihood of $P(C|(s, s'); \theta) = \sigma(\langle \boldsymbol{s}, \boldsymbol{s}' \rangle)$. Then, in the lower bound of the optimal value of $\mathcal{L}_{\mathrm{contrastive}}(\theta)$, $\frac{1}{2}\|\boldsymbol{w}\|^2 \approx \mathrm{KL}(w)$.*

In other words, $\mathrm{KL}(w)$ is encoded in the norm of word embeddings, which construct the sentence embedding. The proof is shown in Section C.2.

## C.1 Assumptions

**Dataset:** Let $\mathcal{S}$ be the training corpus for contrastive learning. When training a sentence encoder with contrastive fine-tuning, the set of sentence pairs $\{(s, s')\}$ is used as the training data, and the sentence encoder is trained with the objective which discriminates whether the sentence pair is a positive example (the pair of semantically similar sentences) or negative example (the pair of semantically dissimilar sentences). For the theoretical analysis, we make the following three reasonable assumptions:

1. An anchor sentence $s$ is sampled from $\mathcal{S}$.

2. Positive exmaple： For semantically similar sentence $s'$ with $s$, $s$ itself is used ($s' = s$).

3. Negative example： For semantically dissimilar sentence $s'$ with $s$, randomly sampled sentence $s'$ from $\mathcal{S}$ is used ($s' \sim \mathcal{S}$).

Assumption 1 is a commonly used setting for the training data of contrastive fine-tuning of sentence encoders (Gao et al., 2021; Chuang et al., 2022, etc.). Assumption 2 considers data augmentation techniques based on perturbations such as token shuffling (Yan et al., 2021) or dropout augmentation (Gao et al., 2021). Noting that these data augmentations do not affect the word frequency distributions, this simplification has little effect on the theory. Assumption 3 considers the most simple way to create dissimilar sentence $s'$ of the anchor sentence $s$, especially in unsupervised settings (Yan et al., 2021; Gao et al., 2021; Chuang et al., 2022, etc.). In typical supervised settings, *hard negatives* are often created with NLI supervision, and considering these settings in the theory is an important future work. Also, for simplicity, we assume the sentence length is fixed to $n$ in the proof ($|s| = |s'| = n$).

**Model:** We make the following three assumptions on contrastive-based sentence encoders (Section 2):

1. The sentence embedding is constructed by the mean pooling (Section 2).

2. The inner product is used to calculate the similarity of sentence embeddings $\boldsymbol{s}$ and $\boldsymbol{s}'$

3. Sentence embedding is not normalized.

For assumption 2, one can also use the inner product instead of cosine similarity, as discussed in Gao et al. (2021), for example. Using the inner product for calculating sentence embedding similarity is an important assumption for the proof [4], and extending our theory to cosine similarity is a challenging and interesting future work. Assumption 3 makes the conclusion of our theory meaningful, which uses the norm of word embedding. Typical contrastive sentence encoders compute sentence embedding without normalization (Reimers and Gurevych, 2019; Gao et al., 2021, etc.).

## C.2 Proof

First, $P(C = 1 | (s, s'); \theta)$ has the following lower bound:

$$
\begin{aligned}
P(C &= 1 | (s, s'); \theta) \\
&= \sigma(\langle \boldsymbol{s}, \boldsymbol{s}' \rangle) &(5) \\
&= \sigma(\sum_{w \in s} \sum_{w' \in s'} \frac{1}{|s||s'|} \langle \boldsymbol{w}, \boldsymbol{w}' \rangle) &(6) \\
&\geq \prod_{w \in s} \prod_{w' \in s'} \sigma(\langle \boldsymbol{w}, \boldsymbol{w}' \rangle)^{\frac{1}{|s||s'|}} &(7) \\
&= \prod_{w \in s} \prod_{w' \in s'} P(C = 1 | (w, w'); \theta)^{\frac{1}{|s||s'|}}. &(8)
\end{aligned}
$$

Here, we used the mean pooling assumption and the property of bilinear form for Equation 6 and Theorem 3.2 in Nantomah (2019) for Equation 7.

Then the objective function $\mathcal{L}_{\text{contrastive}}(\theta)$ of the probabilistic model $P$ has the following approxi-

---

[4]More accurately, our theory requires the similarity function for the sentence embeddings to be a bilinear form. The inner product is a (symmetric) bilinear form, while cosine is not.

mated lower bound:

$$
\mathcal{L}_{\text{contrastive}}(\theta)
$$
$$
= \prod_{(s,s')\sim\text{pos}} P(C=1|(s,s');\theta)
$$
$$
\times \prod_{(s,s')\sim\text{neg}} P(C=0|(s,s');\theta) \tag{9}
$$
$$
\geq \prod_{(s,s')\sim\text{pos}} \left( \prod_{w\in s}\prod_{w'\in s'} P(C=1|(s,s');\theta)^{\frac{1}{|s||s'|}} \right)
$$
$$
\times \prod_{(s,s')\sim\text{neg}} \left( \prod_{w\in s}\prod_{w'\in s'} P(C_s=0|(s,s');\theta)^{\frac{1}{|s||s'|}} \right) \tag{10}
$$
$$
\approx \prod_{(w,w')\sim\text{pos}} P(C=1|(w,w');\theta)^{\frac{1}{n^2}}
$$
$$
\times \prod_{(w,w')\sim\text{neg}} P(C=0|(w,w');\theta)^{\frac{1}{n^2}} \tag{11}
$$
$$
= \left( \prod_{(w,w')\sim\text{pos}} P(C=1|(w,w');\theta) \right.
$$
$$
\left. \times \prod_{(w,w')\sim\text{pos}} P(C=0|(w,w');\theta) \right)^{\frac{1}{n^2}} \tag{12}
$$

Here, Equation 10 follows from Equation 6 to 8, and $|s|=|s'|=n$ is used in Equation 11. Hence, the optimal value for $\mathcal{L}_{\text{contrastive}}$ is bounded as follows:

$$
\max_{\theta} \mathcal{L}_{\text{contrastive}}(\theta)
$$
$$
\geq \max_{\theta} \left( \prod_{(w,w')\sim\text{pos}} P(C_s=1|(w,w');\theta) \right.
$$
$$
\left. \times \prod_{(w,w')\sim\text{pos}} P(C_s=0|(w,w');\theta) \right)^{\frac{1}{n^2}} \tag{13}
$$
$$
\arg\max_{\theta}(13)
$$
$$
= \arg\max_{\theta} \left( \prod_{(w,w')\sim\text{pos}} P(C_s=1|(w,w');\theta) \right.
$$
$$
\left. \times \prod_{(w,w')\sim\text{pos}} P(C_s=0|(w,w');\theta) \right) \tag{14}
$$

Noting that $\arg\max_{\theta}(13) = \arg\max_{\theta}(14)$, Equation 14 corresponds to the objective function of the skip-gram with negative sampling (SGNS) model (Mikolov et al., 2013) with taking the context window as a sentence. In other words, the optimal value of the lower bound (Equation 13) corresponds to the optimal value of the SGNS model (Equation 14).

(i) If $(s,s')$ is a positive example, the words $w'$ in $s'$ can be considered sampled from the distribution of words, co-occurring with $w$ in sentences: $w' \sim P_{\text{pos}}(\cdot|w) = P_{\text{same sent}}(\cdot|w)$. (ii) If $s'$ is a negative example, the word in a negative example can be

considered sampled from the unigram distribution $P(\cdot)$, followed from $s' \sim \mathcal{S}$ and $w' \sim s'$. By using the property of the trained SGNS model shown in Oyama et al. (2023), we have

$$
\frac{1}{2}\|\boldsymbol{w}\|^2 \approx \text{KL}(P_{\text{pos}}(\cdot|w) \parallel P_{\text{neg}}(\cdot|w)) \tag{15}
$$
$$
= \text{KL}(P_{\text{samesent}}(\cdot|w) \parallel P(\cdot)) \quad \square \tag{16}
$$

### C.3 Discussion

Here, we discuss the two implications from Theorem 1.

First, Equation 16 represents the intuition explained in Section 4. That is, the difference of the word frequency distribution of the similar sentence $s'$ and the dissimilar sentence $s'$ after observing the word $w$ in the anchor sentence $s$ is implicitly encoded in $\|\boldsymbol{w}\|$. In other words, the information gain on the word frequency distribution of the similar sentence $s'$ is encoded in $\|\boldsymbol{w}\|$ by observing $w \in s$.

Secondly, the conclusion of our proof that the information gain $\text{KL}(w)$ is encoded in the norm of $\boldsymbol{w}$ justifies the means of quantifying the implicit word weighting of the model using Integrated Gradients (IG) or SHAP to a certain extent. When constructing a sentence embedding by *additive* composition (MEAN pooling), the contribution of each word is approximately determined by the norm of the word embedding (Yokoi et al., 2020). IG and SHAP also *additively* decomposes the contribution of each input feature (input word embedding) to the model output (sentence embedding) (Sundararajan et al., 2017; Lundberg and Lee, 2017). From the perspective of the additive properties, the result that IG and SHAP can capture the contributions implicitly encoded in the norm is natural. To preserve the additive properties of IG and SHAP more properly, further sophisticating the aggregation methods of contributions (Equation 4, 20) is an interesting future work.

## D Quantifying word weighting within sentence encoders with SHAP

In this section, we briefly describe Shapley aadditive explanations (SHAP; Lundberg and Lee, 2017), the feature attribution method introduced in Section 3.1.2, and then describe how to apply SHAP to quantification of word weighting within sentence encoders.

### D.1 Shapley value

SHAP is an extension for XAI of Shapley value (Shapley, 1953), the classic method proposed in the context of cooperative game theory. We first explain Shapley value. Let us consider a cooperative game, where a set of players forms a coalition and gains payoffs. Shapley value is a method to distribute the payoffs gained by cooperation (forming a coalition) fairly among the players. Let $\mathcal{N} := \{1, 2, \ldots, n\}$ be the set of players in the game and $v \colon 2^{\mathcal{N}} \to \mathbb{R}$ be the function that determines the payoff gained based on a subset (coalition) of players. Note that the empty set does not gain payoff, $v(\emptyset) = 0$. To compute the payoff (contribution) $\phi_i$ distributed to the $i$-th player, Shapley value calculates the expectation of the difference of the payoff before and after the $i$-th player joins each of the possible coalitions (all subsets made from permutations of players). Formally, $\phi_i$ is calculated as follows:

$$
\phi_i(v) = \sum_{S \subseteq N \setminus \{i\}} \frac{|S|!(|N| - |S| - 1)!}{N!} \Big( v(S \cup \{i\}) - v(S) \Big).
$$
(17)

Shapley value satisfies some ideal properties; for example, the summation of each calculated contribution $\phi_i$ becomes the payoff of the case where all players join, i.e., $v(N) = \sum_{i \in S} \phi_i(v)$.

### D.2 SHAP

SHAP is an extension of the Shapley value to interpreting machine learning models. Let $f \colon \mathbb{R}^n \to \mathbb{R}$ be a model and $\boldsymbol{X} \in \mathbb{R}^n$ be a certain input. SHAP maps machine learning models to cooperative games: the input $\boldsymbol{X}$ corresponds to the set of players $\mathcal{N}$ and the model $f$ corresponds to the set function $v$ in Equation 17. However, in cooperative games the input (subset of players) is discrete, while in machine learning models the input (vector) is continuous. SHAP fills this discrepancy as follows: the situation "the $i$-th player is not included in the input subset" is mapped to "the $i$-th feature of the input vector is replaced with its expected value." See the original SHAP paper (Lundberg and Lee, 2017) for details.

### D.3 Approximate calculation of SHAP

The exact calculation of Shapley value is computationally expensive (time complexity of $\mathcal{O}(2^N)$) because it calculates all the permutations of the input players (Equation 17). Its extension, SHAP,

is generally calculated through an approximation. One of the typical approximated calculation of SHAP is PartitionSHAP[5] implemented in shap library[6], which hierarchically clusters input features (or words) to decide coalitions, reducing the total number of the permutations in Equation 17. PartitionSHAP is widely used in interpreting NLP models (Mosca et al., 2022; Attanasio et al., 2023; Eksi et al., 2021, etc.); hence, we also use it in our experiments.

### D.4 Application to sentence encoder

When applying SHAP to NLP models, instead of replacing a feature with its expected value, an input word is often replaced with uninformative token (e.g., [MASK] for BERT-based models).[7] For sentence encoders, SHAP calculates the contribution of each input word to the output $m_j(\boldsymbol{W})$ in the form of decomposing $m_j(\boldsymbol{W})$ into a sum:

$$
m_j(\boldsymbol{W}) = \sum_{i=1}^{n} \mathrm{SHAP}_i(\boldsymbol{W}; m_j, \boldsymbol{M}) + f(\boldsymbol{M})
$$
(18)

$$
\boldsymbol{M} = [\boldsymbol{w}_{\texttt{[MASK]}}, \ldots, \boldsymbol{w}_{\texttt{[MASK]}}] \in \mathbb{R}^{n \times d}, \quad (19)
$$

where $\boldsymbol{M}$ denotes a masked input. Then, we can calculate the contribution of $i$-th word to $j$-th element of $\boldsymbol{s}$ by SHAP in the same aggregation as for IG (see Section 3.1.1):

$$
c_i(\boldsymbol{W}; \boldsymbol{m}, \boldsymbol{M}) := \sqrt{\sum_{j=1}^{d} \mathrm{SHAP}_i(\boldsymbol{W}; m_j, \boldsymbol{M})^2}.
$$
(20)

In addition, the word contribution $c_i$ is normalized with respect to the sentence length $n$ same as IG (see Section 3.1.1): $c'_i := c_i / (\frac{1}{n} \sum_{j=1}^{n} c_j)$.

## E Omitted results in Section 5

Tables 2 to 4 are the results of $(\mathrm{IG}, -\log P(w))$, $(\mathrm{SHAP}, \mathrm{KL}(w))$ and $(\mathrm{SHAP}, -\log P(w))$ experiments, respectively, and Figure 6 is the linear regression plots between $-\log P(w)$ and IG.

---

[5] https://shap.readthedocs.io/en/latest/generated/shap.PartitionExplainer.html
[6] https://github.com/shap/shap
[7] The approach replacing inputs with a certain baseline is an instance of Baseline SHAP (BSHAP; Sundararajan and Najmi, 2020).

| Model (pooling) | $R^2 \times 100 \uparrow$ | $\beta \times 100 \uparrow$ |
|---|---|---|
| BERT (CLS) | 0.2 | −0.5 |
| → U.SimCSE-BERT | **37.9** (+37.7) | **10.3** (+10.8) |
| → S.SimCSE-BERT | **36.7** (+36.5) | **10.1** (+10.7) |
| → DiffCSE-BERT | **39.5** (+39.3) | **8.0** (+8.6) |
| BERT (MEAN) | 29.9 | 4.8 |
| → SBERT | 28.4 (−1.5) | **9.8** (+5.0) |
| RoBERTa (CLS) | 1.9 | 3.1 |
| → U.SimCSE-RoBERTa | **36.1** (+34.2) | **12.0** (+8.9) |
| → S.SimCSE-RoBERTa | **36.8** (+34.9) | **14.0** (+11.0) |
| → DiffCSE-RoBERTa | **35.1** (+33.1) | **11.3** (+8.2) |
| RoBERTa (MEAN) | 4.7 | 3.6 |
| → SRoBERTa | **39.2** (+34.5) | **16.6** (+12.9) |
| MPNet (MEAN) | 1.4 | −2.2 |
| all-mpnet-base-v2 | **22.4** (+20.9) | **11.7** (+14.0) |

Table 2: Coefficient of determination ($R^2$) and regression coefficient ($\beta$) of the linear regression of the words' weightings calculated by IG on the self-information $-\log P(w)$ for the STS-B dataset. The $R^2$ and $\beta$ is reported as $R^2 \times 100$ and $\beta \times 100$. The values inside the brackets represent the gain from pre-trained models.

| Model (pooling) | $R^2 \times 100 \uparrow$ | $\beta \times 100 \uparrow$ |
|---|---|---|
| BERT (CLS) | 0.0 | 0.3 |
| → U.SimCSE-BERT | **1.2** (+1.1) | **2.0** (+1.6) |
| → S.SimCSE-BERT | **19.1** (+19.1) | **15.9** (+15.6) |
| → DiffCSE-BERT | **0.7** (+0.7) | **1.3** (+0.9) |
| BERT (MEAN) | 0.5 | −1.3 |
| → SBERT | **10.7** (+10.2) | **18.6** (+19.9) |
| RoBERTa (CLS) | 1.3 | −6.5 |
| → U.SimCSE-RoBERTa | **11.2** (+9.8) | **7.8** (+14.3) |
| → S.SimCSE-RoBERTa | **14.5** (+13.2) | **10.7** (+17.2) |
| → DiffCSE-RoBERTa | **3.2** (+1.8) | **5.0** (+11.5) |
| RoBERTa (MEAN) | 0.0 | 0.3 |
| → SRoBERTa | **14.4** (+14.4) | **13.4** (+13.1) |
| MPNet (MEAN) | 0.7 | 2.8 |
| all-mpnet-base-v2 | **30.8** (+30.1) | **21.9** (+19.0) |

Table 3: Coefficient of determination ($R^2$) and regression coefficient ($\beta$) of the linear regression of the words' weightings calculated by SHAP on the information gain KL for the STS-B dataset. The $R^2$ and $\beta$ is reported as $R^2 \times 100$ and $\beta \times 100$. The values inside the brackets represent the gain from pre-trained models.

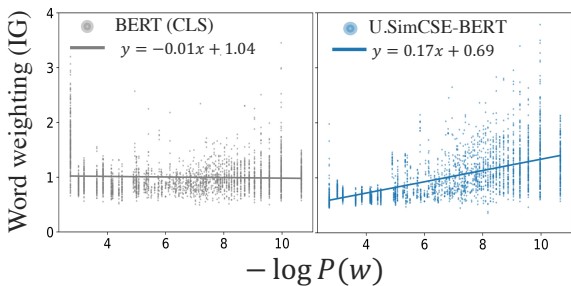

Figure 6: Linear regression plots between $-\log P(w)$ and the word weighting of BERT (CLS) (left) and Unsupervised (U.) SimCSE-BERT (right) using IG. We plotted subsampled 3000 tokens from the tokens with the top 99.5% of small KL values for visibility.

# F  Experiments with Wikipedia, NLI, and STS12 datasets

Here, we conduct experiments with Wikipedia, NLI, and STS12 datasets other than STS-B datasets in Section 5 for generalizability.

## F.1  Experimental setup

The following datasets were used:

- Wikipedia: randomly sampled sentences of Wikipedia used in the SimCSE paper (Gao et al., 2021).[8] We further randomly sampled 3000 sentences.

- NLI: concatenation of SNLI (Bowman et al., 2015) and MNLI (Williams et al., 2018) datasets used in the SimCSE paper (Gao et al., 2021).[9] We further randomly sampled 3000 sentences.

- STS12: randomly sampled 3000 sentences from the test set of STS12 (Agirre et al., 2012) dataset.

We followed the same setting for models and word weighting calculation in Section 5 and we experimented with all the combinations of ({IG, SHAP},{KL$(w)$, $-\log P(w)$}).

## F.2  Results

Tables 5 to 16 are the results of the three datasets. Except for SBERT in the IG experiments[10] and U.SimCSE-BERT/RoBERTa DiffCSE-BERT/RoBERTa in the SHAP experiments, the coefficient of determination $R^2$ and regression coefficient $\beta$ of contrastive-based sentence encoders are higher than pre-trained models across all the three datasets, verifying the consistency with the experiments with STS-B in Section 5 and E. The results suggest that contrastive-based sentence encoders learn to weight each word according to the

[8]https://huggingface.co/datasets/
princeton-nlp/datasets-for-simcse/resolve/main/
wiki1m_for_simcse.txt

[9]https://huggingface.co/datasets/
princeton-nlp/datasets-for-simcse/resolve/main/
nli_for_simcse.csv

[10]$R^2$ does not increase from BERT(MEAN), while $\beta$ indeed increases.

| Model (pooling) | $R^2 \times 100 \uparrow$ | $\beta \times 100 \uparrow$ |
|---|---|---|
| BERT (CLS) | 0.0 | −0.1 |
| → U.SimCSE-BERT | **0.8** (+0.8) | **0.9** (+0.9) |
| → S.SimCSE-BERT | **23.3** (+23.3) | **9.7** (+9.7) |
| → DiffCSE-BERT | **0.5** (+0.5) | **0.6** (+0.7) |
| BERT (MEAN) | 1.4 | −1.2 |
| → SBERT | **14.1** (+12.8) | **11.7** (+13.0) |
| RoBERTa (CLS) | 3.2 | −6.3 |
| → U.SimCSE-RoBERTa | **11.9** (+8.6) | **4.9** (+11.2) |
| → S.SimCSE-RoBERTa | **18.8** (+15.5) | **7.5** (+13.7) |
| → DiffCSE-RoBERTa | 2.1 (−1.1) | **2.5** (+8.8) |
| RoBERTa (MEAN) | 0.1 | −0.7 |
| → SRoBERTa | **20.0** (+19.9) | **9.8** (+10.5) |
| MPNet (MEAN) | 0.6 | 1.4 |
| all-mpnet-base-v2 | **37.0** (+36.4) | **13.2** (+11.7) |

Table 4: Coefficient of determination ($R^2$) and regression coefficient ($\beta$) of the linear regression of the words' weightings calculated by SHAP on the self-information $-\log P(w)$ for the STS-B dataset. The $R^2$ and $\beta$ is reported as $R^2 \times 100$ and $\beta \times 100$. The values inside the brackets represent the gain from pre-trained models.

information-theoretic quantities of words, irrespective of the dataset domains.

| Model (pooling) | $R^2 \times 100 \uparrow$ | $\beta \times 100 \uparrow$ |
|---|---|---|
| BERT (CLS) | 0.6 | 2.2 |
| → U.SimCSE-BERT | **39.3** (+38.7) | **22.7** (+20.4) |
| → S.SimCSE-BERT | **36.5** (+35.9) | **20.7** (+18.5) |
| → DiffCSE-BERT | **40.1** (+39.4) | **16.8** (+14.6) |
| BERT (MEAN) | 32.4 | 10.0 |
| → SBERT | 23.6 (−8.8) | **19.6** (+9.6) |
| RoBERTa (CLS) | 0.8 | 5.5 |
| → U.SimCSE-RoBERTa | **34.1** (+33.2) | **22.3** (+16.8) |
| → S.SimCSE-RoBERTa | **31.9** (+31.1) | **24.3** (+18.8) |
| → DiffCSE-RoBERTa | **31.5** (+30.6) | **20.6** (+15.1) |
| RoBERTa (MEAN) | 1.9 | 5.9 |
| → SRoBERTa | **37.7** (+35.8) | **26.8** (+21.0) |
| MPNet (MEAN) | 0.2 | −2.0 |
| all-mpnet-base-v2 | **23.0** (+22.8) | **25.4** (+27.5) |

Table 5: Coefficient of determination ($R^2$) and regression coefficient ($\beta$) of the linear regression of the words' weightings calculated by IG on the information gain KL for the Wikipedia dataset. The $R^2$ and $\beta$ is reported as $R^2 \times 100$ and $\beta \times 100$. The values inside the brackets represent the gain from pre-trained models.

| Model (pooling) | $R^2 \times 100 \uparrow$ | $\beta \times 100 \uparrow$ |
|---|---|---|
| BERT (CLS) | 0.2 | 0.6 |
| → U.SimCSE-BERT | **41.1** (+40.9) | **11.3** (+10.8) |
| → S.SimCSE-BERT | **39.5** (+39.4) | **10.5** (+10.0) |
| → DiffCSE-BERT | **43.8** (+43.7) | **8.6** (+8.0) |
| BERT (MEAN) | 32.8 | 4.9 |
| → SBERT | 26.9 (−5.9) | **10.2** (+5.3) |
| RoBERTa (CLS) | 0.9 | 3.0 |
| → U.SimCSE-RoBERTa | **36.0** (+35.2) | **12.5** (+9.5) |
| → S.SimCSE-RoBERTa | **35.0** (+34.1) | **13.8** (+10.8) |
| → DiffCSE-RoBERTa | **34.5** (+33.7) | **11.7** (+8.7) |
| RoBERTa (MEAN) | 2.2 | 3.4 |
| → SRoBERTa | **41.3** (+39.1) | **15.3** (+11.9) |
| MPNet (MEAN) | 0.4 | −1.3 |
| all-mpnet-base-v2 | **22.4** (+22.0) | **12.3** (+13.6) |

Table 6: Coefficient of determination ($R^2$) and regression coefficient ($\beta$) of the linear regression of the words' weightings calculated by IG on the self-information $-\log P(w)$ for the Wikipedia dataset. The $R^2$ and $\beta$ is reported as $R^2 \times 100$ and $\beta \times 100$. The values inside the brackets represent the gain from pre-trained models.

| Model (pooling) | $R^2 \times 100 \uparrow$ | $\beta \times 100 \uparrow$ |
|---|---|---|
| BERT (CLS) | 0.2 | −1.6 |
| → U.SimCSE-BERT | **1.3** (+1.1) | **2.9** (+4.5) |
| → S.SimCSE-BERT | **11.4** (+11.2) | **13.4** (+15.0) |
| → DiffCSE-BERT | 0.0 (−0.1) | **0.3** (+1.9) |
| BERT (MEAN) | 0.5 | −1.5 |
| → SBERT | **6.2** (+5.8) | **15.4** (+16.9) |
| RoBERTa (CLS) | 1.7 | −10.3 |
| → U.SimCSE-RoBERTa | **4.7** (+3.0) | **6.9** (+17.2) |
| → S.SimCSE-RoBERTa | **8.0** (+6.3) | **9.3** (+19.6) |
| → DiffCSE-RoBERTa | **2.5** (+0.8) | **5.2** (+15.5) |
| RoBERTa (MEAN) | 0.2 | −1.9 |
| → SRoBERTa | **8.2** (+8.0) | **11.7** (+13.6) |
| MPNet (MEAN) | 0.1 | −1.4 |
| all-mpnet-base-v2 | **21.5** (+21.4) | **20.5** (+22.0) |

Table 7: Coefficient of determination ($R^2$) and regression coefficient ($\beta$) of the linear regression of the words' weightings calculated by SHAP on the information gain KL for the Wikipedia dataset. The $R^2$ and $\beta$ is reported as $R^2 \times 100$ and $\beta \times 100$. The values inside the brackets represent the gain from pre-trained models.

| Model (pooling) | $R^2 \times 100 \uparrow$ | $\beta \times 100 \uparrow$ |
|---|---|---|
| BERT (CLS) | 0.0 | 0.0 |
| → U.SimCSE-BERT | **34.9** (+34.9) | **19.1** (+19.1) |
| → S.SimCSE-BERT | **31.0** (+31.0) | **18.5** (+18.4) |
| → DiffCSE-BERT | **32.8** (+32.8) | **14.2** (+14.2) |
| BERT (MEAN) | 29.5 | 9.3 |
| → SBERT | 24.8 (−4.7) | **18.2** (+8.9) |
| RoBERTa (CLS) | 1.8 | 5.8 |
| → U.SimCSE-RoBERTa | **32.9** (+31.1) | **20.0** (+14.2) |
| → S.SimCSE-RoBERTa | **31.5** (+29.7) | **23.1** (+17.3) |
| → DiffCSE-RoBERTa | **30.2** (+28.4) | **18.8** (+13.0) |
| RoBERTa (MEAN) | 4.2 | 6.5 |
| → SRoBERTa | **34.1** (+29.8) | **28.2** (+21.7) |
| MPNet (MEAN) | 0.6 | −3.0 |
| all-mpnet-base-v2 | **23.0** (+22.4) | **22.4** (+25.4) |

Table 9: Coefficient of determination ($R^2$) and regression coefficient ($\beta$) of the linear regression of the words' weightings calculated by IG on the information gain KL for the NLI dataset. The $R^2$ and $\beta$ is reported as $R^2 \times 100$ and $\beta \times 100$. The values inside the brackets represent the gain from pre-trained models.

| Model (pooling) | $R^2 \times 100 \uparrow$ | $\beta \times 100 \uparrow$ |
|---|---|---|
| BERT (CLS) | 0.3 | −1.1 |
| → U.SimCSE-BERT | **1.4** (+1.1) | **1.5** (+2.6) |
| → S.SimCSE-BERT | **13.8** (+13.5) | **7.2** (+8.3) |
| → DiffCSE-BERT | 0.0 (−0.3) | **0.1** (+1.2) |
| BERT (MEAN) | 0.8 | −1.0 |
| → SBERT | **8.3** (+7.5) | **8.7** (+9.6) |
| RoBERTa (CLS) | 2.8 | −7.1 |
| → U.SimCSE-RoBERTa | **3.9** (+1.2) | **3.4** (+10.5) |
| → S.SimCSE-RoBERTa | **9.1** (+6.3) | **5.4** (+12.5) |
| → DiffCSE-RoBERTa | 1.9 (−0.8) | **2.5** (+9.6) |
| RoBERTa (MEAN) | 0.6 | −2.0 |
| → SRoBERTa | **9.7** (+9.1) | **6.9** (+8.9) |
| MPNet (MEAN) | 0.2 | −1.3 |
| all-mpnet-base-v2 | **23.9** (+23.7) | **10.6** (+11.9) |

Table 8: Coefficient of determination ($R^2$) and regression coefficient ($\beta$) of the linear regression of the words' weightings calculated by SHAP on the self-information $-\log P(w)$ for the Wikipedia dataset. The $R^2$ and $\beta$ is reported as $R^2 \times 100$ and $\beta \times 100$. The values inside the brackets represent the gain from pre-trained models.

| Model (pooling) | $R^2 \times 100 \uparrow$ | $\beta \times 100 \uparrow$ |
|---|---|---|
| BERT (CLS) | 0.3 | −0.6 |
| → U.SimCSE-BERT | **40.1** (+39.9) | **10.2** (+10.8) |
| → S.SimCSE-BERT | **36.0** (+35.7) | **9.9** (+10.5) |
| → DiffCSE-BERT | **39.2** (+38.9) | **7.7** (+8.3) |
| BERT (MEAN) | 34.3 | 5.0 |
| → SBERT | 29.8 (−4.6) | **9.9** (+4.9) |
| RoBERTa (CLS) | 1.9 | 3.3 |
| → U.SimCSE-RoBERTa | **38.8** (+36.9) | **12.1** (+8.8) |
| → S.SimCSE-RoBERTa | **38.6** (+36.7) | **14.2** (+10.9) |
| → DiffCSE-RoBERTa | **36.2** (+34.3) | **11.4** (+8.1) |
| RoBERTa (MEAN) | 5.0 | 3.9 |
| → SRoBERTa | **41.1** (+36.1) | **17.2** (+13.3) |
| MPNet (MEAN) | 1.4 | −2.2 |
| all-mpnet-base-v2 | **24.2** (+22.8) | **11.5** (+13.7) |

Table 10: Coefficient of determination ($R^2$) and regression coefficient ($\beta$) of the linear regression of the words' weightings calculated by IG on the self-information $-\log P(w)$ for the NLI dataset. The $R^2$ and $\beta$ is reported as $R^2 \times 100$ and $\beta \times 100$. The values inside the brackets represent the gain from pre-trained models.

| Model (pooling) | $R^2 \times 100 \uparrow$ | $\beta \times 100 \uparrow$ |
|---|---|---|
| BERT (CLS) | 0.2 | 1.6 |
| → U.SimCSE-BERT | **1.2** (+0.9) | **2.1** (+0.5) |
| → S.SimCSE-BERT | **18.4** (+18.2) | **17.5** (+15.8) |
| → DiffCSE-BERT | **1.8** (+1.6) | **2.0** (+0.4) |
| BERT (MEAN) | 0.6 | −1.5 |
| → SBERT | **10.7** (+10.1) | **20.3** (+21.8) |
| RoBERTa (CLS) | 1.3 | −7.2 |
| → U.SimCSE-RoBERTa | **8.7** (+7.3) | **7.6** (+14.8) |
| → S.SimCSE-RoBERTa | **15.1** (+13.8) | **11.8** (+19.0) |
| → DiffCSE-RoBERTa | **3.2** (+1.9) | **5.1** (+12.3) |
| RoBERTa (MEAN) | 0.0 | 0.2 |
| → SRoBERTa | **14.1** (+14.1) | **14.8** (+14.6) |
| MPNet (MEAN) | 0.8 | 3.1 |
| all-mpnet-base-v2 | **30.9** (+30.1) | **23.3** (+20.2) |

Table 11: Coefficient of determination ($R^2$) and regression coefficient ($\beta$) of the linear regression of the words' weightings calculated by SHAP on the information gain KL for the NLI dataset. The $R^2$ and $\beta$ is reported as $R^2 \times 100$ and $\beta \times 100$. The values inside the brackets represent the gain from pre-trained models.

| Model (pooling) | $R^2 \times 100 \uparrow$ | $\beta \times 100 \uparrow$ |
|---|---|---|
| BERT (CLS) | 0.0 | −0.4 |
| → U.SimCSE-BERT | **37.9** (+37.9) | **19.3** (+19.8) |
| → S.SimCSE-BERT | **31.5** (+31.5) | **17.3** (+17.8) |
| → DiffCSE-BERT | **37.1** (+37.0) | **14.9** (+15.3) |
| BERT (MEAN) | 28.6 | 9.2 |
| → SBERT | 24.0 (−4.6) | **17.3** (+8.0) |
| RoBERTa (CLS) | 1.6 | 5.3 |
| → U.SimCSE-RoBERTa | **36.6** (+35.1) | **21.1** (+15.8) |
| → S.SimCSE-RoBERTa | **33.9** (+32.3) | **23.5** (+18.2) |
| → DiffCSE-RoBERTa | **35.6** (+34.1) | **20.2** (+14.9) |
| RoBERTa (MEAN) | 4.1 | 6.4 |
| → SRoBERTa | **38.3** (+34.1) | **28.6** (+22.2) |
| MPNet (MEAN) | 0.8 | −3.2 |
| all-mpnet-base-v2 | **24.1** (+23.3) | **23.1** (+26.3) |

Table 13: Coefficient of determination ($R^2$) and regression coefficient ($\beta$) of the linear regression of the words' weightings calculated by IG on the information gain KL for the STS12 dataset. The $R^2$ and $\beta$ is reported as $R^2 \times 100$ and $\beta \times 100$. The values inside the brackets represent the gain from pre-trained models.

| Model (pooling) | $R^2 \times 100 \uparrow$ | $\beta \times 100 \uparrow$ |
|---|---|---|
| BERT (CLS) | 0.2 | 0.8 |
| → U.SimCSE-BERT | **0.9** (+0.7) | **0.9** (+0.2) |
| → S.SimCSE-BERT | **23.3** (+23.1) | **9.8** (+9.0) |
| → DiffCSE-BERT | **1.9** (+1.7) | **1.0** (+0.3) |
| BERT (MEAN) | 1.5 | −1.1 |
| → SBERT | **14.8** (+13.3) | **11.8** (+12.9) |
| RoBERTa (CLS) | 2.7 | −5.7 |
| → U.SimCSE-RoBERTa | **9.4** (+6.7) | **4.4** (+10.1) |
| → S.SimCSE-RoBERTa | **19.5** (+16.9) | **7.4** (+13.1) |
| → DiffCSE-RoBERTa | 2.4 (−0.3) | **2.5** (+8.1) |
| RoBERTa (MEAN) | 0.0 | −0.4 |
| → SRoBERTa | **19.4** (+19.4) | **9.7** (+10.1) |
| MPNet (MEAN) | 0.8 | 1.5 |
| all-mpnet-base-v2 | **38.0** (+37.3) | **12.9** (+11.4) |

Table 12: Coefficient of determination ($R^2$) and regression coefficient ($\beta$) of the linear regression of the words' weightings calculated by SHAP on the self-information $-\log P(w)$ for the NLI dataset. The $R^2$ and $\beta$ is reported as $R^2 \times 100$ and $\beta \times 100$. The values inside the brackets represent the gain from pre-trained models.

| Model (pooling) | $R^2 \times 100 \uparrow$ | $\beta \times 100 \uparrow$ |
|---|---|---|
| BERT (CLS) | 0.4 | −0.7 |
| → U.SimCSE-BERT | **39.9** (+39.6) | **10.4** (+11.1) |
| → S.SimCSE-BERT | **33.6** (+33.2) | **9.4** (+10.1) |
| → DiffCSE-BERT | **40.7** (+40.3) | **8.2** (+8.9) |
| BERT (MEAN) | 29.6 | 4.9 |
| → SBERT | 27.0 (−2.6) | **9.6** (+4.7) |
| RoBERTa (CLS) | 1.3 | 2.7 |
| → U.SimCSE-RoBERTa | **38.0** (+36.7) | **12.3** (+9.6) |
| → S.SimCSE-RoBERTa | **36.7** (+35.4) | **14.0** (+11.3) |
| → DiffCSE-RoBERTa | **37.6** (+36.3) | **11.9** (+9.2) |
| RoBERTa (MEAN) | 3.9 | 3.6 |
| → SRoBERTa | **40.4** (+36.5) | **16.8** (+13.3) |
| MPNet (MEAN) | 1.3 | −2.2 |
| all-mpnet-base-v2 | **23.3** (+21.9) | **11.9** (+14.1) |

Table 14: Coefficient of determination ($R^2$) and regression coefficient ($\beta$) of the linear regression of the words' weightings calculated by IG on the self-information $-\log P(w)$ for the STS12 dataset. The $R^2$ and $\beta$ is reported as $R^2 \times 100$ and $\beta \times 100$. The values inside the brackets represent the gain from pre-trained models.

| Model (pooling) | $R^2 \times 100 \uparrow$ | $\beta \times 100 \uparrow$ |
|---|---|---|
| BERT (CLS) | 0.0 | −0.6 |
| → U.SimCSE-BERT | **0.9** (+0.8) | **1.9** (+2.4) |
| → S.SimCSE-BERT | **21.1** (+21.1) | **17.0** (+17.5) |
| → DiffCSE-BERT | **0.4** (+0.4) | **1.0** (+1.6) |
| BERT (MEAN) | 1.2 | −2.3 |
| → SBERT | **11.8** (+10.6) | **20.0** (+22.3) |
| RoBERTa (CLS) | 2.4 | −9.6 |
| → U.SimCSE-RoBERTa | **11.0** (+8.7) | **8.6** (+18.2) |
| → S.SimCSE-RoBERTa | **17.2** (+14.9) | **12.9** (+22.5) |
| → DiffCSE-RoBERTa | **2.8** (+0.5) | **5.0** (+14.6) |
| RoBERTa (MEAN) | 0.1 | −1.2 |
| → SRoBERTa | **19.1** (+19.0) | **16.7** (+17.9) |
| MPNet (MEAN) | 0.4 | 2.5 |
| all-mpnet-base-v2 | **34.1** (+33.6) | **23.6** (+21.1) |

Table 15: Coefficient of determination ($R^2$) and regression coefficient ($\beta$) of the linear regression of the words' weightings calculated by SHAP on the information gain KL for the STS12 dataset. The $R^2$ and $\beta$ is reported as $R^2 \times 100$ and $\beta \times 100$. The values inside the brackets represent the gain from pre-trained models.

| Model (pooling) | $R^2 \times 100 \uparrow$ | $\beta \times 100 \uparrow$ |
|---|---|---|
| BERT (CLS) | 0.1 | −0.5 |
| → U.SimCSE-BERT | **0.7** (+0.6) | **0.8** (+1.3) |
| → S.SimCSE-BERT | **22.7** (+22.7) | **9.2** (+9.7) |
| → DiffCSE-BERT | **0.2** (+0.2) | **0.4** (+0.9) |
| BERT (MEAN) | 2.0 | −1.5 |
| → SBERT | **13.8** (+11.8) | **11.3** (+12.9) |
| RoBERTa (CLS) | 3.8 | −7.0 |
| → U.SimCSE-RoBERTa | **10.7** (+6.9) | **4.8** (+11.8) |
| → S.SimCSE-RoBERTa | **19.0** (+15.2) | **7.7** (+14.7) |
| → DiffCSE-RoBERTa | 2.0 (−1.8) | **2.4** (+9.4) |
| RoBERTa (MEAN) | 0.3 | −1.4 |
| → SRoBERTa | **22.1** (+21.8) | **10.3** (+11.7) |
| MPNet (MEAN) | 0.3 | 1.0 |
| all-mpnet-base-v2 | **36.7** (+36.5) | **12.8** (+11.8) |

Table 16: Coefficient of determination ($R^2$) and regression coefficient ($\beta$) of the linear regression of the words' weightings calculated by SHAP on the self-information $-\log P(w)$ for the STS12 dataset. The $R^2$ and $\beta$ is reported as $R^2 \times 100$ and $\beta \times 100$. The values inside the brackets represent the gain from pre-trained models.