# OpenReview forum: "Contrastive Learning-based Sentence Encoders Implicitly Weight Informative Words"
_EMNLP/2023/Conference — EMNLP 2023 Findings_

### Official Review · Reviewer_3ibg · 2023-08-07

**Soundness:** 4

**Excitement:**

4: Strong: This paper deepens the understanding of some phenomenon or lowers the barriers to an existing research direction.

**Paper Topic And Main Contributions:**

This paper studies the interpretability of the contrastive learning-based sentence encoder.

More specifically, the authors use the integrated gradients method to analyze the SimCSE contrastive learning model and investigate how the encoders internally weigh input words to encode a sentence. Their experiments reveal that contrastive learning induces the encoders to weight input words according to information-theoretic quantities, which are somewhat similar to previous sentence embedding methods.

**Reasons To Accept:**


1. The method is intuitive and makes sense.
2. The experimental setup is reasonable and well-executed. The results support their claim.
3. The writing and math presentation is clear and easy to follow.

**Reasons To Reject:**

N/A

**Reproducibility:**

3: Could reproduce the results with some difficulty. The settings of parameters are underspecified or subjectively determined; the training/evaluation data are not widely available.

**Reviewer Confidence:**

3: Pretty sure, but there's a chance I missed something. Although I have a good feel for this area in general, I did not carefully check the paper's details, e.g., the math, experimental design, or novelty.

---

> ### Author Rebuttal · Authors · 2023-08-29
>
> Thanks for the detailed and thoughtful feedback! We appreciate the positive feedback on the intuitiveness of our method, the conciseness of our experimental setting, writing, and math presentation. We will brush up our work more on the camera-ready version!

---

### Official Review · Reviewer_aaka · 2023-08-08

**Soundness:** 2

**Excitement:**

3: Ambivalent: It has merits (e.g., it reports state-of-the-art results, the idea is nice), but there are key weaknesses (e.g., it describes incremental work), and it can significantly benefit from another round of revision. However, I won't object to accepting it if my co-reviewers champion it.

**Paper Topic And Main Contributions:**

This paper endeavors to unravel the underlying mechanisms of contrastive methods used in sentence representation learning.
It does so by introducing nuanced concepts grounded in information theory, such as self-information and information gain.
Specifically, the proposed methodology leverages the Integrated Gradients (IG) technique to assess the implicit weightings attributed to individual words within a sentence, as calculated by various sentence encoders.
Subsequently, the relationship between these computed weights and key information-theoretic quantities (namely, self-information and information gain) is examined. This analysis facilitates a comparative evaluation of sentence embeddings derived from different sentence encoders, shedding light on their distinct characteristics and behavior.

**Reasons To Accept:**

- Attempted to explain the inner workings of contrastive methods for sentence representation learning, with the introduction of simple but intuitive information-theoretic concepts such as self-information and information gain.

**Reasons To Reject:**

- The paper lacks a compelling rationale for the specific use of the Integrated Gradients method to determine the implicit weights of individual words, leaving unanswered questions about the potential outcomes or advantages of employing alternative approaches.
- Given a wide array of contrastive methods for sentence representation learning that extend SimCSE, a simple comparison between the original BERT & RoBERTa models and their SimCSE versions may be insufficient for drawing a robust and well-grounded conclusion.


**Reproducibility:**

4: Could mostly reproduce the results, but there may be some variation because of sample variance or minor variations in their interpretation of the protocol or method.

**Reviewer Confidence:**

4: Quite sure. I tried to check the important points carefully. It's unlikely, though conceivable, that I missed something that should affect my ratings.

---

> ### Author Rebuttal · Authors · 2023-08-29
>
> We are thankful for your comprehensive review! We appreciate the positive feedback on bridging contrastive learning for sentence embedding and information-theoretic quantities.
> We also appreciate the helpful feedback on the experimental setting of IG and other possible methods besides the models to be analyzed.
>
> # Additional experiment with other contrastive-based sentence encoders
> Thank you for the constructive feedback on the varieties of models to be analyzed!
> We fully agree that our experiments are limited to BERT & RoBERTa and their SimCSE versions, and adding more models to be analyzed would help us understand more about the contrastive learning of sentence encoders (ll.307-313).
>
> To validate the generalizability of our claim, we conducted additional experiments with contrastive-based sentence encoders other than SimCSE/SBERT.
> We choose DiffCSE ([Chuang et al., 2022](https://aclanthology.org/2022.naacl-main.311/)), an extension of the SimCSE model, incorporating an ELECTRA-like training objective.
> We also select [all-mpnet-base-v2](https://huggingface.co/sentence-transformers/all-mpnet-base-v2) ([Reimers and Gurevych, 2019](https://aclanthology.org/D19-1410/)), one of the SoTA sentence transformer models. It is fine-tuned from MPNet ([Song et al., 2020](https://proceedings.neurips.cc/paper/2020/hash/c3a690be93aa602ee2dc0ccab5b7b67e-Abstract.html)) using contrastive learning with various labeled pair sentences.
>
> As a result, we confirmed that the coefficient of determination $R^2$ and regression coefficient $\beta$ of contrastive-based sentence encoders are higher than those of models before contrastive fine-tuning for all contrastive-based models (denoted as ✔), supporting the generalizability of our claim.
> We will include these results in the camera-ready version! Again, thank you for the constructive feedback.
>
> | Model (pooling)         | $R^2$ w.r.t. $-\log{P(w)}$ ↑ | $\beta$ w.r.t. $-\log{P(w)}$ ↑ | $R^2$ w.r.t. $\mathrm{KL}(w)$ ↑ | $\beta$ w.r.t. $\mathrm{KL}(w)$ ↑ |
> |----------------------|-----------------|----------------|--------------|-------------|
> | BERT(CLS)            | 0.066           | 0.022          | 0.079        | 0.110       |
> | → DiffCSE(CLS)            | 0.393 ✔︎          | 0.080 ✔︎         | 0.334 ✔︎       | 0.153 ✔︎      |
> | MPNet(CLS)           | 0.016           | -0.028         | 0.006        | -0.030      |
> | MPNet(MEAN)          | 0.0142          | -0.022         | 0.006        | -0.026      |
> | →　all-mpnet-base-v2(MEAN) | 0.224 ✔︎          | 0.117 ✔︎         | 0.216 ✔︎       | 0.210 ✔︎      |
>
> # Reasons for using Integrated Gradients / Additional experiments with SHAP
> Thank you for pointing out the reasons for using Integrated Gradients (IG) and the potential usage of other methods!
>
> We chose IG because IG is a widely used technique among feature attribution methods for differentiable neural networks (ll.111-118).
> IG satisfies desirable axioms, such as sensitivities, while other gradient-based methods often violate these axioms ([Sundararajan et al., 2017](https://proceedings.mlr.press/v70/sundararajan17a.html)).
>
> Besides gradient-based feature attribution methods, SHAP ([Lundberg and Lee, 2017](https://papers.nips.cc/paper_files/paper/2017/hash/8a20a8621978632d76c43dfd28b67767-Abstract.html)), which is based on the Shapley Value used in game theory ([Lloyd S, 1953](https://www.jstor.org/stable/j.ctt1b9x1zv.24)), is also a widely used feature attribution method in the NLP field as well ([Mosca et al., 2022](https://aclanthology.org/2022.coling-1.406/)).
>
> To validate that our claim also holds for methods other than IG, we conducted experiments using SHAP. In addition to models analyzed in the paper, we added DiffCSE ([Chuang et al., 2022](https://aclanthology.org/2022.naacl-main.311/)) which is an extension of SimCSE, and [all-mpnet-base-v2](https://huggingface.co/sentence-transformers/all-mpnet-base-v2) ([Reimers and Gurevych, 2019](https://aclanthology.org/D19-1410/)), which is one of SoTA sentence transformer models based on MPNet ([Song et al., 2020](https://proceedings.neurips.cc/paper/2020/hash/c3a690be93aa602ee2dc0ccab5b7b67e-Abstract.html)) for the analysis.
>
> As a result, we confirmed that the $R^2$ and $\beta$ of contrastive-based sentence encoders are higher than those of models before contrastive fine-tuning for almost all experimental combinations (denoted as ✔), supporting the generalizability of our claim.
>
> | Model (pooling)         | $R^2$ w.r.t. $-\log{P(w)}$ ↑ | $\beta$ w.r.t. $-\log{P(w)}$ ↑ | $R^2$ w.r.t. $\mathrm{KL}(w)$ ↑ | $\beta$ w.r.t. $\mathrm{KL}(w)$ ↑ |
> |-------------------------|----------------------|---------------------|-------------------|------------------|
> | BERT(CLS)               | 0.006                | -0.009              | 0.002             | -0.010           |
> | → Unsup. SimCSE-BERT(CLS)    | 0.006                | 0.008 ✔︎              | 0.010 ✔︎             | 0.018 ✔︎           |
> | → Sup. SimCSE-BERT(CLS)     | 0.227 ✔︎                 | 0.094 ✔︎              | 0.186 ✔︎            | 0.154 ✔︎           |
> | → DiffCSE-BERT(CLS)          | 0.006                | 0.006 ✔︎              | 0.008 ✔︎             | 0.013 ✔︎           |
> | BERT(MEAN)              | 0.014                | -0.012              | 0.005             | -0.013           |
> | → SBERT(MEAN)           | 0.131 ✔︎                | 0.117 ✔︎              | 0.107 ✔︎             | 0.186 ✔︎           |
> | RoBERTa(CLS)            | 0.029                | -0.056              | 0.012             | -0.057           |
> | → Unsup. SimCSE-RoBERTa(CLS) | 0.119 ✔︎                 | 0.050 ✔︎              | 0.113 ✔︎              | 0.078 ✔︎           |
> | → Sup. SimCSE-RoBERTa(CLS)   | 0.189 ✔︎                | 0.074 ✔︎              | 0.146 ✔︎              | 0.106 ✔︎           |
> | RoBERTa(MEAN)           | 0.000                | -0.007              | 0.000             | 0.003            |
> | → SRoBERTa(MEAN)              | 0.200 ✔︎                 | 0.098 ✔︎              | 0.144 ✔︎             | 0.134 ✔︎           |
> | mpnet(CLS)              | 0.000                | 0.004               | 0.002             | 0.014            |
> | mpnet(MEAN)             | 0.005                | 0.014               | 0.007             | 0.028            |
> | →　all-mpnet-base-v2(MEAN)    | 0.370 ✔︎                 | 0.132 ✔︎              | 0.308 ✔︎             | 0.219 ✔︎           |

---

### Official Review · Reviewer_4dqe · 2023-08-10

**Soundness:** 2

**Excitement:**

3: Ambivalent: It has merits (e.g., it reports state-of-the-art results, the idea is nice), but there are key weaknesses (e.g., it describes incremental work), and it can significantly benefit from another round of revision. However, I won't object to accepting it if my co-reviewers champion it.

**Paper Topic And Main Contributions:**

Main contribution: it tries to explain how contrastive learning gives weights to input words in a given sentence
- It compares a word's contribution to contrastive learning's sentence encoding with two information-theoretic quantities: self-information & information gain.
- It computes a word's contribution to sentence encoding using Integrated Gradients.

**Reasons To Accept:**

- Easy idea to follow.
- Paper easy to read. (Short paper)
- (Perhaps) first paper that tackles to understand why contrastive learning works the ways it works

**Reasons To Reject:**

- Qualitative analysis is very limited and ambiguous. ("cutting" is clear, but how about "onion"? For 'onion', -log P(w) is lower than SimCSE-BERT. )
- What do we know about the inner working of contrastive learning by comparing word's contribution (from IG) against self-information and information gain? The authors need to take one more step to interpret the results further.

**Reproducibility:**

3: Could reproduce the results with some difficulty. The settings of parameters are underspecified or subjectively determined; the training/evaluation data are not widely available.

**Reviewer Confidence:**

2: Willing to defend my evaluation, but it is fairly likely that I missed some details, didn't understand some central points, or can't be sure about the novelty of the work.

---

> ### Author Rebuttal · Authors · 2023-08-29
>
> Thank you for the insightful review! We appreciate the positive feedback on our idea and tackling demystifying the contrastive learning of sentence encoder. We also appreciate the constructive feedback on the insights of our work.
>
> # Qualitative analysis of word weighting of the models
> Thank you for commenting about the qualitative analysis!
>
> The "onion" in Figure 3 is actually a good example from the information theory perspective. Our writing might have confused you, so let us briefly explain here.
>
> Our hypothesis throughout the paper is that contrastive-based sentence encoders weight each word according to information-theoretic quantities $-\log{P(w)}$ or $\mathrm{KL}(w)$. Thus, in Figure 3, we would like to see if blue (before contrastive fine-tuning) has a different word weighting trend from gray lines (information-theoretic quantities) and if orange (after contrastive fine-tuning) has a similar trend with gray (information-theoretic quantities) lines.
>
> Through this perspective with Figure 3, we can observe the following trends that support our hypothesis:
> - Blue (before contrastive fine-tuning) assigns high weight to low informative words such as *is* and punctuation.
> - However, orange (after contrastive fine-tuning) assigns high weight to informative words such as *woman*, *cutting* and *onion* when constructing sentence embedding.
>
> A similar trend is also observed in Figure 25 in the appendix.
>
> Besides, this interesting phenomenon of contrastive sentence encoders' weighting informative words is observed in the whole corpus (Section 4.2), not only in the specific sentence in Figure 3. The increase of $R^2$ and $\beta$ in Table 1 quantitatively explains that contrastive-based sentence encoders weight informative words when constructing sentence embedding.
>
> Based on your feedback, we will add the discussion here to the caption of Figure 3 in the camera-ready version.
>
> # Insights of the inner mechanism of contrastive learning for sentence embedding
> Thank you for your important perspective!
> To summarize our work, we estimated how each input word contributes to the output (sentence embedding) in sentence encoders trained with contrastive learning and compared it with information theoretic quantities. Our results show the macro-level characteristic that "sentence encoders learn to focus on informative words via contrastive fine-tuning."
>
> We believe our insight further advances the understanding of contrastive learning for sentence embedding from previous micro-level analysis, which focuses on the inner mechanism of representations: [Gao et al., 2021](https://aclanthology.org/2021.emnlp-main.552/) showed that contrastive learning induces representations for semantically similar sentences to be distributed closer and vice versa in sentence embedding space.
> Our work further attempts to demystify how contrastive learning works from the word-level contribution in sentence encoders and association with information theory.
>
> Our insight is also helpful for understanding the practical success of contrastive learning. For example, our insight that the sentence encoders learn IDF-like word weighting ($-\log{P(w)}$) through contrastive fine-tuning provides the understanding of why contrastive fine-tuning is also effective in tasks such as passage retrieval where the similarity of the topics should be considered.
>
> In the camera-ready version, we will add one paragraph about the point discussed here using an additional page. Again, thank you very much for the important suggestion! If anything is unclear, let us discuss it in the discussion phase!

---

### Official Review · Reviewer_un9Z · 2023-08-11

**Soundness:** 3

**Excitement:**

3: Ambivalent: It has merits (e.g., it reports state-of-the-art results, the idea is nice), but there are key weaknesses (e.g., it describes incremental work), and it can significantly benefit from another round of revision. However, I won't object to accepting it if my co-reviewers champion it.

**Paper Topic And Main Contributions:**

The paper explores the weighting of input words in encoders for contrastive learning.
The mathematical background of the method is very strong and very detailed.


The experimental results are limited.

**Questions For The Authors:**

- Is there any paper for the state given in lines 40-41 on the first page?

- Is there any extra method to quantify the word weighting addition to Integrated Gradients? If yes, what is the reason of choosing this method?

- What is your conclusion about the higher scores of sBERT/SRoBERTA compared to SimCSE BERT/SimCSE RoBERTa?

**Reasons To Accept:**

The paper presents solid proof of contrastive learning word weighting by information gain.

**Reasons To Reject:**

- The experimental results by using only STS-B are very limited. There are several STS datasets, especially it would be nice to see the results of STS12 which has lower scores compared to other datasets.

**Reproducibility:**

3: Could reproduce the results with some difficulty. The settings of parameters are underspecified or subjectively determined; the training/evaluation data are not widely available.

**Reviewer Confidence:**

2: Willing to defend my evaluation, but it is fairly likely that I missed some details, didn't understand some central points, or can't be sure about the novelty of the work.

**Typos Grammar Style And Presentation Improvements:**

line 159 discriimination
line 181 ward w
line 222 OLS -> the full version of OLS ?

---

> ### Author Rebuttal · Authors · 2023-08-29
>
> Thank you for the thoughtful review! We appreciate the positive feedback on our mathematical proof of why contrastive learning induces word weighting by information gain.
> Also, thank you for pointing out important points in the experimental settings.
>
> # Additional experiments using STS-12 and other datasets to validate our claim.
> Thank you for suggesting this!
> We fully agree that our result should be confirmed with other datasets than STS-B (ll.302-307).
> STS-12, which differs in domains from STS-B, would be helpful to confirm the generalizability.
>
> ## Experimental setting
> Following your suggestion, we conducted experiments using STS-12 dataset for BERT-based models to validate the generalizability of our claim.
> We used test set of STS-12 to calculate the word weighting of the model (IG), self-information and information gain.
>
> We also experimented with Wikipedia and NLI datasets whose domains differ from STS-B/STS-12 for further validation.
> We followed the same setting with SimCSE ([Gao et al., 2021](https://aclanthology.org/2021.emnlp-main.552/)):
> - Wikipedia: randomly sampled 30k sentences.
> - NLI: concatenation of SNLI ([Bowman et al., 2015](https://aclanthology.org/D15-1075/)) and MNLI([Williams et al., 2018](https://aclanthology.org/N18-1101/)) datasets.
>
> Whole datasets were used to calculate self-information and information gain.
> For calculating IG, we randomly sampled the 1500 sentences from each dataset.
>
> ## Results
> The experimental results show that coefficient of determination $R^2$ and regression coefficient $\beta$ for contrastive-finetuned encoders are higher than those for pre-trained models across almost all datasets, models, and information-theoretic quantities combinations (denoted as ✔︎).
> These results are consistent with the results of STS-B, validating our claim that contrastive learning induces encoders to weight words according to $-\log{P(w)}$ and $\mathrm{KL}(w)$. We will include these results in the camera-ready version as well. Again, thank you for the constructive feedback.
>
> ###  Result for STS-12
> | Model (pooling)         | $R^2$ w.r.t. $-\log{P(w)}$ ↑ | $\beta$ w.r.t. $-\log{P(w)}$ ↑ | $R^2$ w.r.t. $\mathrm{KL}(w)$ ↑ | $\beta$ w.r.t. $\mathrm{KL}(w)$ ↑ |
> |----------------------|-----------------|----------------|--------------|-------------|
> | BERT(CLS)            | 0.059           | 0.023          | 0.065        | 0.046       |
> | → Unsup. SimCSE-BERT(CLS) | 0.397 ✔︎           | 0.102 ✔︎         | 0.377 ✔︎        | 0.190 ✔︎      |
> | → Sup. SimCSE-BERT(CLS)   | 0.326 ✔︎           | 0.091 ✔︎         | 0.306 ✔︎      | 0.168 ✔︎      |
> | BERT(MEAN)           | 0.296           | 0.049          | 0.286        | 0.092       |
> | → SBERT(MEAN)              | 0.270           | 0.096 ✔︎         | 0.240        | 0.173 ✔︎      |
>
> ### Result for Wikipedia
> | Model (pooling)         | $R^2$ w.r.t. $-\log{P(w)}$ ↑ | $\beta$ w.r.t. $-\log{P(w)}$ ↑ | $R^2$ w.r.t. $\mathrm{KL}(w)$ ↑ | $\beta$ w.r.t. $\mathrm{KL}(w)$ ↑ |
> |----------------------|-----------------|----------------|--------------|-------------|
> | BERT(CLS)            | 0.090           | 0.0314         | 0.128        | 0.248       |
> | → Unsup. SimCSE-BERT(CLS) | 0.427 ✔︎           | 0.106 ✔︎         | 0.409 ✔︎       | 0.686 ✔︎      |
> | → Sup. SimCSE-BERT(CLS)   | 0.399 ✔︎          | 0.096 ✔︎         | 0.352 ✔︎       | 0.598 ✔︎      |
> | BERT(MEAN)           | 0.333           | 0.046          | 0.333        | 0.304       |
> | → SBERT(MEAN)              | 0.269           | 0.095 ✔︎        | 0.213        | 0.556 ✔︎      |
>
> ### Result for NLI
> | Model (pooling)         | $R^2$ w.r.t. $-\log{P(w)}$ ↑ | $\beta$ w.r.t. $-\log{P(w)}$ ↑ | $R^2$ w.r.t. $\mathrm{KL}(w)$ ↑ | $\beta$ w.r.t. $\mathrm{KL}(w)$ ↑ |
> |----------------------|-----------------|----------------|--------------|-------------|
> | BERT(CLS)            | 0.066           | 0.022          | 0.079        | 0.110       |
> | → Unsup. SimCSE-BERT(CLS) | 0.403 ✔︎          | 0.092 ✔︎         | 0.413 ✔︎       | 0.426 ✔︎      |
> | → Sup. SimCSE-BERT(CLS)   | 0.347 ✔︎          | 0.087 ✔︎         | 0.325 ✔︎       | 0.387 ✔︎      |
> | BERT(MEAN)           | 0.330           | 0.045          | 0.302        | 0.198       |
> | → SBERT(MEAN)              | 0.296           | 0.090 ✔︎         | 0.266        | 0.391 ✔︎      |
>
>
> # Explicit word weighting techniques to improve sentence embedding quality
> Thank you for the question!
> Weighting static word embeddings by the quantities based on the inverse of word frequency, such as TF-IDF ([Arroyo-Fernández et al., 2019](https://linkinghub.elsevier.com/retrieve/pii/S0885230817302887)) and smoothed inverse frequency (SIF; [Arora et al., 2017](https://openreview.net/forum?id=SyK00v5xx)), are strong baselines, outperforming more complicated sentence encoders based on LSTM or RNN (ll.21-26). In the era of BERT-based sentence encoder, it is reported that weighting each word of SBERT model further improves the embedding qualities ([Wang and Kuo, 2020](https://ieeexplore.ieee.org/document/9140343)).
>
> # Reasons for using Integrated Gradients / Additional experiments with SHAP
> Thank you for the question regarding Integrated Gradients (IG) and other posibble methods!
>
> We chose IG because IG is a widely used technique among feature attribution methods for differentiable neural networks (ll.111-118).
> IG satisfies desirable axioms, such as sensitivities, while other gradient-based methods often violate these axioms ([Sundararajan et al., 2017](https://proceedings.mlr.press/v70/sundararajan17a.html)).
>
> Besides gradient-based feature attribution methods, SHAP ([Lundberg and Lee, 2017](https://papers.nips.cc/paper_files/paper/2017/hash/8a20a8621978632d76c43dfd28b67767-Abstract.html)), which is based on the Shapley Value used in game theory ([Lloyd S, 1953](https://www.jstor.org/stable/j.ctt1b9x1zv.24)), is also a widely used feature attribution method in the NLP field as well ([Mosca et al., 2022](https://aclanthology.org/2022.coling-1.406/)).
>
> To validate that our claim also holds for methods other than IG, we conducted experiments using SHAP. In addition to models analyzed in the paper, we added DiffCSE ([Chuang et al., 2022](https://aclanthology.org/2022.naacl-main.311/)) which is an extension of SimCSE, and [all-mpnet-base-v2](https://huggingface.co/sentence-transformers/all-mpnet-base-v2) ([Reimers and Gurevych, 2019](https://aclanthology.org/D19-1410/)), which is one of SoTA sentence transformer models based on MPNet ([Song et al., 2020](https://proceedings.neurips.cc/paper/2020/hash/c3a690be93aa602ee2dc0ccab5b7b67e-Abstract.html)) for the analysis.
>
> As a result, we confirmed that the $R^2$ and $\beta$ of contrastive-based sentence encoders are higher than those of models before contrastive fine-tuning for almost all experimental combinations (denoted as ✔), supporting the generalizability of our claim.
>
> | Model (pooling)         | $R^2$ w.r.t. $-\log{P(w)}$ ↑ | $\beta$ w.r.t. $-\log{P(w)}$ ↑ | $R^2$ w.r.t. $\mathrm{KL}(w)$ ↑ | $\beta$ w.r.t. $\mathrm{KL}(w)$ ↑ |
> |-------------------------|----------------------|---------------------|-------------------|------------------|
> | BERT(CLS)               | 0.006                | -0.009              | 0.002             | -0.010           |
> | → Unsup. SimCSE-BERT(CLS)    | 0.006                | 0.008 ✔︎              | 0.010 ✔︎             | 0.018 ✔︎           |
> | → Sup. SimCSE-BERT(CLS)     | 0.227 ✔︎                 | 0.094 ✔︎              | 0.186 ✔︎            | 0.154 ✔︎           |
> | → DiffCSE-BERT(CLS)          | 0.006                | 0.006 ✔︎              | 0.008 ✔︎             | 0.013 ✔︎           |
> | BERT(MEAN)              | 0.014                | -0.012              | 0.005             | -0.013           |
> | → SBERT(MEAN)           | 0.131 ✔︎                | 0.117 ✔︎              | 0.107 ✔︎             | 0.186 ✔︎           |
> | RoBERTa(CLS)            | 0.029                | -0.056              | 0.012             | -0.057           |
> | → Unsup. SimCSE-RoBERTa(CLS) | 0.119 ✔︎                 | 0.050 ✔︎              | 0.113 ✔︎              | 0.078 ✔︎           |
> | → Sup. SimCSE-RoBERTa(CLS)   | 0.189 ✔︎                | 0.074 ✔︎              | 0.146 ✔︎              | 0.106 ✔︎           |
> | RoBERTa(MEAN)           | 0.000                | -0.007              | 0.000             | 0.003            |
> | → SRoBERTa(MEAN)              | 0.200 ✔︎                 | 0.098 ✔︎              | 0.144 ✔︎             | 0.134 ✔︎           |
> | mpnet(CLS)              | 0.000                | 0.004               | 0.002             | 0.014            |
> | mpnet(MEAN)             | 0.005                | 0.014               | 0.007             | 0.028            |
> | →　all-mpnet-base-v2(MEAN)    | 0.370 ✔︎                 | 0.132 ✔︎              | 0.308 ✔︎             | 0.219 ✔︎           |
>
>
>
> # Scores of SBERT/SRoBERTA compared to SimCSE BERT/SimCSE RoBERTa
> Thank you for the question! In Table 1 in the paper, the $R^2$ values are SBERT > SimCSE-BERT while SRoBERTa < SimCSE-RoBERTa. This result suggests that the score of $R^2$, which indicates how strong IG and $-\log{P(w)}$ or ${\mathrm{KL(w)}}$ are linearly associated, is not a perfect proxy for how good the sentence encoder is.
> There might be more appropriate quantities that can explain the word weighting of the model (IG) other than $-\log{P(w)}$ and ${\mathrm{KL(w)}}$, which we have yet to be aware of, and it is left to future work (ll.288-290).

---

### Meta-Review · Area_Chair_pDtF · 2023-09-13

**Recommendation:** 4

**Metareview:**

This paper studies the interpretability of the contrastive learning-based sentence encoder. More specifically, the authors use the integrated gradients method to analyze the SimCSE contrastive learning model and investigate how the encoders internally weigh input words to encode a sentence. The experiments reveal that contrastive learning induces the encoders to weight input words according to information-theoretic quantities, which are somewhat similar to previous sentence embedding methods.

I did not find in the reviews any strong reason to reject the paper except that its most important contribution is theoretical rather than empirical. To me, this is not an actual limitation and I think this paper can make a good contribution to the EMNLP audiance.

---

### Decision · Program_Chairs · 2023-10-07

**Decision:**

Accept-Findings

**Comment:**

This paper studies the interpretability of the contrastive learning-based sentence encoder. More specifically, the authors use the integrated gradients method to analyze the SimCSE contrastive learning model and investigate how the encoders internally weigh input words to encode a sentence. The experiments reveal that contrastive learning induces the encoders to weight input words according to information-theoretic quantities, which are somewhat similar to previous sentence embedding methods.

I did not find in the reviews any strong reason to reject the paper except that its most important contribution is theoretical rather than empirical. To me, this is not an actual limitation and I think this paper can make a good contribution to the EMNLP audiance.